# The Use of Draught Animals in Rural Labour

**DOI:** 10.3390/ani11092683

**Published:** 2021-09-13

**Authors:** Daniel Mota-Rojas, Ada Braghieri, Adolfo Álvarez-Macías, Francesco Serrapica, Efrén Ramírez-Bribiesca, Rosy Cruz-Monterrosa, Felicia Masucci, Patricia Mora-Medina, Fabio Napolitano

**Affiliations:** 1Neurophysiology, Behavior and Animal Welfare Assessment, DPAA, Campus Xochimilco, Universidad Autónoma Metropolitana, México City 04960, Mexico; dmota100@yahoo.com.mx (D.M.-R.); aalvarez@correo.xoc.uam.mx (A.Á.-M.); 2Scuola di Scienze Agrarie, Forestali, Alimentari ed Ambientali, Università degli Studi della Basilicata, 85100 Potenza, PZ, Italy; ada.braghieri@unibas.it; 3Dipartimento di Agraria, Università degli Studi di Napoli Federico II, 80055 Portici, NA, Italy; francesco.serrapica@unina.it (F.S.); masucci@unina.it (F.M.); 4Livestock Program, Colegio de Postgraduados, Texcoco 56230, Mexico; efrenrb@colpos.mx; 5Food Science Department, Universidad Autónoma Metropolitana-Unidad Lerma, Lerma 52172, Mexico; r.cruz@correo.ler.uam.mx; 6Livestock Department, Universidad Nacional Autónoma de México (UNAM), FESC, Mexico City 54715, Mexico; mormed2001@yahoo.com.mx

**Keywords:** bovids, equids, cattle, buffaloes, horses, donkeys, draught animals, animal welfare, rural activities, greenhouse gas emission

## Abstract

**Simple Summary:**

Although mechanisation has markedly reduced animal labour demand in agriculture, draught animals are still used in small production units located on terrains that do not favour agriculture mechanisation. Especially in Africa, Latin America, and Asia, they represent one of the main sources of sustenance for thousands of families who utilise animal labour in numerous agricultural tasks, such as ploughing and harvesting, as well as means of transport and hauling. Depending on the geographic area, the species involved are equids (horses, donkeys, and mules) and bovids (buffaloes and cattle). Draught animals proved to be sustainable in terms of global warming and the use of non-renewable energy as compared with agricultural machinery. However, critical points are the quality of human–animal interaction, due to the close contact between animals and humans while working, and the welfare of draught animals when transported and slaughtered, due to the high prevalence of injuries they suffer when subjected to these practices. Therefore, their use should be promoted in rural marginal areas where only low investments are usually feasible, and the energy of the animals can be obtained at a low cost by feeding them harvest residues and by-products.

**Abstract:**

This study discusses scientific findings on the use of draught animals such as equids (i.e., horses, mules, and donkeys) and bovids (i.e., cattle and water buffaloes) in rural labours. Relevant peer-reviewed literature published between 1980 and 2021 was retrieved from CAB Abstracts, PubMed, ISI Web of Knowledge, and Scopus databases. Although animals were used to produce draught power since their domestication and are still being used for this purpose, mechanisation has markedly reduced animal labour demand in agriculture. However, the process was uneven across continents according to economic constraints, and draught animals are currently concentrated in small production units located on terrains that do not favour agriculture mechanisation in Africa, Latin America, and Asia. Generally, equids can work at rates similar to those of bovids or faster but can sustain the work for shorter periods of time. In addition, buffaloes possess tough hooves and resistance to disease that make them suitable for working in wetlands and clay soils. Draught animals allow a marked reduction of both GHG emissions and non-renewable energy consumption as compared with agricultural machinery. In addition, they may allow obtaining profits from otherwise non-usable lands. Therefore, their use should be promoted in rural areas where low investments are usually the only ones feasible, and the energy of the animals can be obtained at a low cost by feeding them harvest residues and by-products. However, more attention should be paid to the quality of human–animal interactions—due to the close contact between animals and humans while working—and to the welfare of draught animals when transported and slaughtered—due to the high prevalence of injuries they suffer when subjected to these practices.

## 1. Introduction

Animals have been involved in the sociocultural and economic evolution of societies. Most significantly they participated in the process of utilization of natural resources to produce food and work in rural societies and, more broadly, they promoted the generation of wealth [1,2].

The most common species still used for transport and draught are equids, such as horses, mules, and donkeys, and bovids, such as cattle, including oxen, and buffaloes [1,3,4,5]. The type of orography and soil may also prioritise the use of one species over another; for example, flat, dry lands are usually worked by donkeys, horses, or even camels, while oxen are preferred in mountainous areas and buffaloes are used for clay soils and areas subject to flooding [4,6].

Although no up-to-date data on the number of draught animals working in the world are available (in 1982 the Food and Agriculture Organization estimated that 400 million working animals were used), draught animals persist as a primary source of livelihood for thousands of families in developing countries, and rural families, often including children, participate actively in the management and handling practices associated with draught animals, exerting a significant influence on their welfare [7].

From a normative perspective, humans are obliged to ensure the quality of life of these animals [8,9], not only because of the value they generate in terms of work, food, and economic income, but also in terms of ethical concerns [1]. These considerations should provide sufficient incentive to focus on the topic of the welfare of draught animals [1,10].

Our capacity to perceive and understand the needs of other living beings has been a key feature of the process of providing protection for farm animals based on the knowledge of their physiological characteristics and behavioural needs, which are specific for draught animals. For these reasons, it is important to improve the attitude towards draught animals to provide an acceptable level of welfare to these animals in agreement with the principles of sustainable development [2]. Accordingly, the World Organisation for Animal Health has been promoting the welfare of equids, including those used as draught animals, through reports concerning good practices encouraging good feeding, good health, provision of shelter, and appropriate workload when used as means of transport [11].

The use of draught animals lightens human labour in cultivation, loading, and transport. However, the participation of draught animals to agricultural activities can also contribute to the farm economy [12]. The added value of livestock in agricultural systems through both their traction activities and production of manure may well be just as significant—or even more so—than that of the meat and/or milk they produce. In East Africa, for example, the first two aspects total 42% versus 38% for meat and 17% for milk. In Central Africa, in contrast, the resources of animal traction and fertiliser are of only marginal value, whereas in the other four regions of the continent, it is described as “important” [12]. Although the study that provided these figures is not recent, it indicates the importance of the contribution of draught animals, which is so often underestimated [12].

It is from this perspective that this study discusses scientific findings on the use of draught animals such as equids (i.e., horses, mules, and donkeys) and bovids (i.e., oxen and water buffaloes) in rural labours, the relations among draught animals, farmers, and rural families, the contribution of these animals to global warming, the quality of life of pack and traction animals, and their welfare conditions at the end of their working lives.

## 2. Methods

Relevant peer-reviewed literature published between 1980 and 2021 was retrieved from CAB Abstracts, PubMed, ISI Web of Knowledge, and Scopus databases. The search was based on the following search terms: draught animals, pack animals, animal energy, draught power, working animals, animal welfare, human–animal relationship, feeding, economy, buffalo, *Bubalus bubalis*, oxen, equids, horse, donkey, mule, yoke, attaching animals to vehicles, land preparation, rural activities, greenhouse gas emission. Boolean strings were constructed based on meaningful combinations of search terms and operators. Search terms were repeatedly refined through a multistep process in several trials to make sure that the most effective search terms were used [13]. The authors also reviewed the sources cited in the identified articles to broaden the search and add relevant materials. The number of publications retrieved and then used per topic are depicted in Figure 1.

## 3. Current Use

Since prehistoric times, when hunters-gatherers changed their status to sedentary and settled, humans benefited from the muscular strength of large, domesticated animals to enhance their capacity for work in agricultural production and transport [14]. Throughout human history, agricultural production has undergone a series of changes that respond to a growing demand for foods and the need to adapt to the conditions imposed by the sociocultural and economic environment. In this process, the use of domesticated species with the capacity to perform agricultural work has been an integral element of the constantly evolving productive models [15]. Currently, in Latin America and Africa, both donkey and horses are commonly used. In particular, in Africa, along with mechanisation in large scale farms and manual labour, which is still predominant in family farms, draught animals have been introduced in the past century and are currently used in small scale farms [16]. However, in some circumstances, mules are preferred to horses as pack animals because they can adapt to low-quality feed and do not need to consume as much water. In addition, their hide is thicker and less susceptible to suffering sores caused by saddles or other riding and harnessing equipment [16]. In general, unless related to local traditions, equids are preferred over bovids when soils are lighter and draught needs are correspondingly lower.

The specific characteristics of buffaloes are the strong and large hooves, flexible foot joints that enhance their work performance and allow higher efficiency in clayey agricultural soils subject to flooding, which demand greater traction effort, whereas camelids are preferred in Asian dry environments [17]. The main limitation of water buffaloes is that they are slower at work [18]. Nevertheless, they possess unique attributes, such as strength and tough hooves, and resistance to disease [19] that gives important advantages for working in wetlands and flooded, heavy, clay soils, where the efficiency of agricultural machinery tends to be limited [20]. Their life cycle as draught animals can last for as much as two decades [4,20]. The use of buffaloes as draught animals is widespread in Asia, particularly in lowland areas of Bangladesh, India, Philippines, Indonesia, and Vietnam, and Latin America. They are widely used as draught animal all year round [4] with higher performances as compared with cattle. For example, buffaloes can carry heavier loads and travel longer distances for longer periods than oxen; they can also work during the night with shorter rest periods [18]. It has been estimated that buffaloes can drag up to about six times their body weight, but normally they carry between 1.5 and 2.0 tonnes (three to four times their own weight). In South-East Asia, the water buffalo predominantly provides traction for work in rice fields where the main breeds are Manda and Paralakhemundi [21]. Other breeds include the Carabao in Vietnam, Binhu in China, Kundhi in Pakistan, Nili-Ravi in India and the Egyptian buffalo, all of which have characteristics favourable for performing tasks that require traction [4]. In the Philippines, the buffalo is currently used as a draught animal in agriculture, mainly by small producers. The main crops grown in Vietnam are rice, sugarcane, corn, peanuts, and soya, where the buffalo has long been the main energy source in agriculture as these animals are utilised for transport and for preparing fields, while their dung provides fresh organic fertiliser for crops [21]. The water buffalo is a docile animal, a behavioural characteristic that allows farmers to train it for activities that require the use of ploughs, rakes, sledges, and wagons. Buffaloes can start their productive life at an earlier age of 2–3 years, depending on the quality of their diet, while conventional cattle are usually ready to work at 3–5 years of age [22]. Then, the buffalo can continue to work productively and efficiently for as many as 15 to 20 years. At the end of their productive life, when sent for slaughter, they may still weigh around 380 kg [4] and their meat represents an important food source [23].

Currently, the use of cattle as draught animals is quite common in Asia and Latin America, while in Africa these animals are used to pull carts or ploughs and their diffusion is only limited by dry environmental conditions and diseases [17]. In general, cattle and buffaloes are more appropriate than equids when the draught requirements are higher.

Mechanisation markedly reduced the labour demand in agriculture [24]. However, the process was uneven across continents according to economic constraints [25] and caused a sharp dichotomy between developed countries, with prevailing agriculture mechanisation, and developing countries, where the use of animal labour is still prevalent. For example, in India, Dikshit and Birthal [26] noticed a reduction of more than 20 million draught animals from 1972 to 2003, whereas in China a decline of the number of draught animals per hectare from 0.6 in the mid-1990s to 0.1 in 2012 was compensated by a concomitant increase of agricultural machine use rate from 25% to more than 50% [27]. However, animal labour is still widely used by small-scale producers and/or by farmers located in marginal areas with scarce resources who integrate these animals into their agriculture activities and field labours both as a means of transport and as draught animals [15,16]. In particular, the use of draught animals is concentrated among small production units located on terrains that do not favour agriculture mechanisation [17]. Some countries in Latin America and Southern Asia satisfy over 35% of the energy utilised in agricultural labours with animals raised for these tasks [28]. In India, there are around 70 million draught animals, which plough approximately 65% of the cultivated land of country [22]. The energy provided by animals constitutes a resource accessible to small producers that allows them to increase the efficiency and productivity of their operations. Available evidence shows that farmers using animal energy generate higher economic incomes than those who perform activities manually because they achieve a higher efficiency and effectiveness through the entire chain of the productive processes [17]. Although agricultural production systems on a larger scale, or with a higher degree of intensification, have gradually moved to mechanisation [29], animal traction can replace numerous forms of labour that some farmers are still performing manually, particularly in Africa [27]. Wilson [30] estimated that in 2000 animal labour was used in about 50% of the global land.

## 4. Production Efficiency and Economic Impact

Numerous studies compared the work produced by different animals (reviewed by Pearson and Vall [31]) and generally concluded that equids can work at rates similar to those of bovids or faster but can sustain the work for shorter periods of time. Conversely, Rahman et al. [32] compared the yield and variable costs of wheat cultivation utilising animal versus mechanical energy. According to the authors, the wheat yields were 2.65 and 2.57 tons/hectare, respectively, for mechanised and animal energy, while the variable costs were significantly higher for animal energy.

Both animal and mechanical energy have positive impacts on production as compared with manual labour. However, mechanisation has proven to be more efficient even after taking into account the substantial investments required for inputs such as infrastructure and equipment (fixed costs), and the need to hire or train personnel to operate the equipment [27]. However, for small and medium producers only low investments are feasible, and the energy of animals can be obtained at a low cost by feeding harvest residues and by-products to the animals (variable costs) and acquiring the appropriate equipment, such as harnesses and related gear, which in most cases represent the only fixed costs [28]. In addition, mechanisation may be impaired in hilly and mountainous regions, which, instead, may be more easily accessed by draft animals [27]. For example, a study conducted on small-scale agricultural enterprises in Mexico [6], showed that draught animals allowed a total average gross income of USD 490.78 per year and production unit. More recently in a study conducted in Ethiopia, the average net profitability (considering income generated by livestock minus the costs of keeping them) of households owning and using equids was estimated to be USD 330 per year [33]. Similarly, in Asia, species such as buffalo have become pillars of agricultural activities [4] thanks to their efficiency that, under certain conditions, has been described as higher to that of tractors, as a small herd of buffaloes may generate benefits for farmers such as reduction of expenditures on fuel and machinery maintenance [21], while potentially reducing the emission of greenhouse gases (GHG) [34]. In the urban environment, working animals are used as a means of transport (for people, goods, and water, among others), in the construction industry and even in the tourism sector [33].

Certainly, economic and productive efficiency improves with the incorporation of mechanised processes to replace animal energy. However, issues concerning sustainable production [32] should be also considered when addressing the issue of draught animals. In particular, the use of draught animals may contribute to an increase in the amount of organic matter in the soils by adding liquid and solid excreta at no cost to the farmer. Both represent a source of biofertilisation for crops [18]. Farmers using mechanical energy, in contrast, usually purchase fertilisers and other agrochemicals to be added to the soil and maintain its fertility [35]. Therefore, animals may produce draught power following good agroecological practices that support adequate soil management while supporting the biological processes involved in soil–plant–animal relations, whereas mechanical power relies on fossil fuels and has a much larger impact on soil compactness [36].

## 5. Greenhouse Gas Emission and Sustainability

The main greenhouse gases (GHG) emitted from draught animals are carbon dioxide, methane, and nitrous oxide. The amount of GHG emitted by draught animals changes according to the species. Bovids show higher levels of emissions due to ruminal and enteric fermentations as compared to the monogastric equids [34]. However, when expressed in relation to body weight, differences between bovids and equids tend to decrease [37].

Mechanisation has markedly increased agriculture productivity and labour efficiency. However, only few studies have assessed the effect of the shift from animals to machinery in terms of GHG emissions. Aguillera et al. [34] showed that the percent contribution of draught animals to global warming decreased from 100% in 1900 to minor levels in current years concomitantly with a marked increase of production and mechanisation levels. For example, in Southern Europe, a sharp reduction of GHG emitted by draught animals occurred from 1970 onward, while a sharp increase of the total agricultural emission of GHG was also observed. These authors suggested that a mixed scenario with the use of draught animals in marginal areas for the provision of draught power in slope low accessibility lands may contribute to the reduction of the GHG emissions in agriculture due to the lower amount produced by the animals as compared with conventional agricultural machines. In addition, the amount of GHG estimated produced by draught animals should be further reduced by taking into account the co-products and ecosystem services that many draught animals often provide [38,39]. Furthermore, the energy produced by draught animals can be considered renewable because they can be replaced according to the needs while also using inputs such as by-products and feeds that are often inedible by humans [26]. Dikshit and Birthal [26] estimated a higher consumption of 19.34 million tonnes of fossil non-renewable fuel per year to replace all draught animals working in India with a corresponding estimated production of 6.14 million tonnes of CO_2_. Cerutti et al. [40] using a life cycle assessment approach estimated a marked reduction of both GHG emission and non-renewable energy consumption by using equids for either forest logging operations or seedbed preparation as compared with agricultural machinery. In some African and Latin American countries, the water buffalo is being used as a means of transport in the palm agribusiness, slightly reducing the enormous environmental impact of this industrial agriculture activity [41]. In contrast, in countries such as Indonesia, Thailand, and Myanmar, the water buffalo is gradually being replaced in agriculture by automated traction equipment. Therefore, in these countries, buffaloes are currently and predominantly used for meat and milk production. In addition, the constantly increasing drought due to climate change may decrease the ability of buffaloes to cope and work [42,43].

## 6. The Welfare of Draught Animals

### 6.1. Human-Animal Interaction

The main factors negatively affecting the welfare of draught animals are listed in Table 1.

The interaction between humans and animals is a keystone of the domestication and subsequent use of farm animals in agriculture, and a good human-animal relationship is fundamental to improving the welfare of both humans and animals [50]. However, the welfare of draught animals may be jeopardised by inadequate handling practices—such as heavy loads, excessive number of working hours, long distance transport in inappropriate condition prior to slaughter—while preventive technical and medical measures should be implemented to improve the welfare and performance of the animals used in agricultural and family labours [44]. The quality of the interaction acquires even greater importance when it comes to draught animals because the proximity is higher, the time of contact is longer, and they both should work in synchrony to achieve predetermined results [47]. Therefore, improper handling can increase the fear perceived by the animals and induce reactions that may jeopardise both human and animal welfare with higher risks of injuries and the onset of negative emotional states [51]. Humans may be perceived as a threat by the animals, while workers may develop a negative attitude towards the animals. A good level of animal welfare does not mean simply keeping animals healthy and well-fed, it requires the achievement of a good emotional state. In addition, striking the animals, twisting their tails, or prodding them with pointed instruments may produce injuries that include abrasions, sores, haematomas, and scarring, among other serious forms of damage [52]. Swann [45] observed a low level of welfare in draught horses in developing countries after applying a behavioural assessment protocol and examining the physical conditions of the animals. The approach revealed that, because the animals were kept in poor welfare conditions, they showed little interest in their surroundings and barely interacted with humans. These results were attributed to the fact that many animals suffered injuries that caused chronic pain. Heat stress and chronic fatigue were identified as additional factors that limited levels of animal welfare. These studies indicate that training of the handlers may play a fundamental role to increase the welfare of draught animals. Moreover, it has been shown that good human–animal relations may have positive effects on the performance of the animals in agricultural labour [50].

### 6.2. Health of Draught Animals

Draught animals may suffer fatigue, malnutrition, and diseases that prevent them from achieving optimum performances in their labours and maintain satisfactory welfare levels. Hyperthermia may be a major factor causing the onset of fatigue [53]. The oxygen saturation of blood may be reduced while the depletion of glycogen reserves may also play a role in the onset of fatigue, as well as the accumulation of lactate in the blood occurring in animals subjected to heavy workloads [54].

Current evaluations of the welfare of draught horses are based on parameters of health and behaviour, including lameness, injuries, and general body conditions [55]. Around 90% of equines employed as pack animals were lame due to the volume and weight of the loads they were forced to carry [45]. Lameness can worsen the condition of animals that may already be suffering health problems, such as malnutrition or dehydration, among others. The transport of heavy loads on roads with an uneven and hard surface can also play a significant role, which may be exacerbated by lack of foot care, inappropriate use of harnesses and other gear and, in general, absence of measures to prevent the deterioration of welfare of draught animals [45].

Recently, Attia et al. [56] noted that all of 120 work donkeys examined in Egypt suffered parasitic infections. These authors verified the presence of at least one parasite in each animal. *Cylicocyclus asini* was the most frequent, identified in 91.7% of cases, followed by *Cyathostomum* spp. in 83.3% of the animals. These results clearly show that it is essential to include deworming procedures in the care of these animals to improve their health in the short term, whereas preventive measures should be taken in the medium term.

With respect to bovines employed as draught animals, a study designed to evaluate the management of oxen based on several parameters-health, feeding, housing, workload, care—showed that 78% of farmers did not provide regular veterinary care [46]. As to feeding, 66% of the animals tested did not receive the amount or quality of feed they required, and most of the horses (76%) were kept tied up in the open. However, the most critical aspect was the use of yokes (harnesses), which caused sores and lesions, largely due to poor maintenance and inappropriate use of the equipment, as 99% of the farmers never cleaned the gear, and most of them did not use cushions to prevent abrasion. Similarly, rusty ploughs and inadequate storing at the end of agricultural labours exacerbated these kinds of injuries and led to visible infections in some of the animals [46].

In the Birbhum district of West Bengal (India) the health of 810 buffaloes randomly selected from herds of draught animals was examined and it was found that several buffaloes reported wounds caused by ploughs while working (16.96%), yoke gall (9.82%), and leg traumas (15.17%). In addition, due to overwork, buffaloes showed hypoglycaemia and nutritional deficits (vitamins and minerals) (15.17%), which were accentuated in summer (17.44%) [18]. Another important cause of animal mistreatment was the duration of the working day. Estimates suggest this may last from 6 h to as many as 10 h a day on 56% of the farms. Farmers, however, may extend the working time in periods of high demand of agricultural labour, such as planting and harvesting, even though this practice may have marked and negative effects on the welfare of the animals [44]. Work tools and equipment can also affect welfare, as they may be inadequately attached when the animals are prepared for working in agriculture, hauling, or transport, thus becoming a frequent cause of skin lesions. Saddles, for example, can produce sores due to friction, chafing, and excessive pressure when fastened too tightly [57].

Most of these deleterious effects can be mitigated by using the appropriate equipment and proper maintenance and fastening procedures. When using animals for hauling or transport, special attention must be paid to balancing the weight of the cart or wagon and to correctly harnessing the animals. Handlers must strive to ensure that draught power is transmitted by pushing a collar with the least amount of pressure and drag possible on the animal’s back, neck, and shoulders [57]. The design of, and materials utilised in, the gear used with animals in rural labours should generate an effective transmission of the effort to achieve optimal working conditions and prevent injuries and irritation negatively affecting the welfare of the animals.

When the working efficiency of draught animals decreases, most of them are sent for slaughter. Differently from productive categories, draught animals often reach this stage when they are fatigued or even exhausted [54]. Therefore, before arriving at the abattoir they are potentially more exposed to suffer falls, blows, and slipping during transport, or when forced to climb up or down ramps, or to pass through doors during loading and unloading into and out of transport vehicles. Alam et al. [58] observed that 99% of the buffaloes and bovines presented visible lesions over almost the entire body when they reached their destination. It was determined that the lesions were associated with the conditions of transport and inadequate handling practices by the personnel in charge of loading and unloading. Injuries such as ulcerations, lacerations, bleeding, dislocations, fractures, and muscle bruising, as well as broken tails, horns, and snouts can be observed at the slaughterhouse. Most of these injuries can be attributed to the excessive number of animals in the vehicle and the insufficient space allowance during transport [59]. Bad transport, loading and unloading conditions are also a cause of emotional stress exacerbating the already compromised welfare state of the animals while also increasing pain perception [59].

A common management practice imposed to draught animals is nose piercing, which is performed inserting a hot iron rod through the septum and then passing a rope through the hole. This practice can cause tearing or chafing injuries to the nostrils, particularly during transport, loading and unloading when the rope is more often used to lead the animal [58]. According to Alam et al. [60], 47% of the animals suffered ulcerations and lacerations caused by ropes or the metal hoop in their noses. In addition, tail lesions, most likely caused by tail twisting, which is often used to encourage movement, occurred in 51% of cattle and 15% of buffaloes. These differences may be due to the different anatomical characteristics of the two species. These results clearly show the level of vulnerability of the animals when subjected to transport. That study included analyses of blood biomarkers (total plasma proteins, serum sodium, plasma glucose, serum unesterified fatty acids, and serum creatine kinase) and every single indicator was above normal levels in both buffaloes and cattle [60].

## 7. Feeding Draught Animals

The working capacity of draught animals depends largely on the availability and use efficiency of feed nutrients [61]. However, draught animals are usually empirically fed, neglecting the nutritional requirements that working activities imply, particularly in the rural areas where high-quality forage availability is rather limited and seasonally fluctuating. In most developing countries, draught animals feeding management practices have naturally evolved to maximize the efficiency of locally available resources, including poor quality feeds and crop residues, such as the stubble and straw of cereals, which are characteristically low in nitrogen and high in fibre [61,62,63].

In hilly and upland areas of the Philippines and Mexico, maize stover supplied half of the diets of draught ruminants and equids for 6 months of the year [64,65]. In sugar-producing areas, cane tops were used to replace natural herbage during the dry season [66]. The Asian draught buffaloes’ diets were based on weeds, green rice straw, maize stover, cassava tops, leaves of trees, banana, and bamboo [65,67,68]. In Tropical America, draught animals are fed with sorghum, wheat and corn residues supplemented with agro-industrial wastes, primarily cotton seed hulls, and corn cobs [69,70]. Mosses, lichens, and hardy mountain grasses were largely used for feeding animals used for rural labour at high Andes and subarctic regions [71,72]. However, all these feeds can be low in nutritional quality, mainly at the late of dry and at the start of the rain season, when animals are required to perform the most work [31]. Consequently, efficiency at work can be impaired since the animals are forced to over-exert to perform the normal activity, and rapidly become fatigued [52,73].

As a long-term effect, feeding working animals with unbalanced diets can lead to poor health status and metabolic disorders, such as an increasing body calcium removal from bone by oxalic acid, particularly abundant in cereal crops by-products [52,74]. Thus, supplying draught animals with high-quality feeds to meet work requirements is a relevant issue. The major needs of the working animals are for high energy-yielding nutrients [73]. As suggested by studies on nitrogen balance, work does not appear to affect protein metabolism in the animals specialized in draught purposes, such as adult male, castrate, or non-productive female [75,76,77]. Similarly, there is no clinical evidence suggesting specific amino acids and vitamin deficiencies because of over prolonged working periods [78]. Moreover, under normal working conditions, draught animals do not require extra minerals other than those feeding standards recommendation, except for minerals strictly related to energy muscle supply (i.e., Ca, Mn, P) and the increased sweating and salivation (mainly, Cl and Na) [73,79].

### 7.1. Energy Requirements

Glucose and long-chain fatty acids (LCFA) have been highlighted as critical nutrients to sustain working muscle activity and to restores body fat during resting periods [73]. In particular, studies of substrate utilization by equid’s and ruminant’s muscles have shown that glucose oxidation is obligatory in muscles, and LCFA are the major fuel [48]. Equine muscles have a high capacity of glycogen storage, which provides considerable glucogenic reserves and the most circulating glucose is absorbed directly from the gut [80]. In ruminants, instead, LCFA are thought to play an important role as energy providers because both the availability of circulating glucose and glycogen reserves are largely dependents on hepatic gluconeogenesis of volatile fatty acids [81]. Therefore, mobilization and oxidation of LCFA, as well as body ketones, increase as work continues, determining a draught capacity reduction and weight losses if dietary supplies or body reserves are insufficient to meet of high-energy substrates availability [81,82].

In mature male working animals, needs of energy sources can be partially offset by catabolizing amino acids, which may be redirect to gluconeogenesis or, as observed in buffaloes [83], directly oxidized in working muscles [78,84]. In addition, in lactating animals the demand for glucose and glucogenic precursors can compete with that for other productive functions, with potential detrimental effects on milk production and reproduction performances as glucose is not only the energy source for muscles activity but also a major substrate for milk lactose synthesis, as well as an essential source of energy for reproductive processes [85,86,87,88]. Similarly, fatty acids are the precursors of milk fat but also play a central role into neuroendocrine activities and the synthesis of reproductive hormones [89,90,91]. Therefore, when lactating cows work, a marked competition for these metabolites by several metabolic pathways occur. In particular, Zerbini et al. [92] observed a plasma glucose concentration reduction during working hours, indicating substantial drainage of glucose from the blood flow to the muscles. At same time, plasma concentration of non-esterified fatty acids increased 1.5-fold over the course of working hours. Similarly, Matthewman et al. [85] highlighted that exercise increase plasma concentration of ß-hydroxybutyrate and free fatty acids, while concentration of glucose, magnesium, and inorganic phosphorous decreased. A significant increase of urinary nitrogen excretion was found in lactating cows at work [92], confirming the results observed by Pieterson and Teleni [93], who reported increased plasma urea and urinary nitrogen in working buffalo cows. These increments may be related to plasma glucose reduction [92], which may induce the cows to remove glucose precursors arising from protein turn-over processes, thereby decreasing the possibility to capture nitrogen in protein re-synthesis [94]. Both yield and milk quality may be influenced by the working activity. Over a period of 3 years, Gemeda et al. [92] observed an average 10% milk production decrement in working compared to non-working cows. Similar results were observed by Zerbini et al. [95]. Work can also affect milk quality as observed on Hereford x Holstein cows by Matthewman et al. [85]. These authors noted that exercise temporarily reduced milk protein and lactose content. In general, working cows delay ovarian activity resumption, show an increased length of the oestrous cycle and, consequently, low conception rates and longer calving intervals [96,97,98]. In particular, comparative studies on working and non-working lactating cows showed 1-day conception delay for each working day [99].

Energy expenditure of draught animals varies according to the work performed. Factors such as the travelled distance, weight of the load, surface characteristics over which the animals move, and the duration of the working day can affect the extent of extra daily energy requirements over maintenance [31,99]. Net energy costs of various activities occurring during work have been laboratory-estimated for several draught animal species and used, if the practical working circumstances are similar, to meet the daily energy requirements. Energy costs for pull and carry loads are relatively constant and, apart from the pack saddle design and weight of the load, strictly related to individual animal species and their live weight, rather than to other external factors [100,101,102]. To carry one kg of load, donkeys spend, on average, 1.7 J/m [102,103], while for Brahman cattle and buffaloes the estimated energy costs reach 2.6 and 4.2 J/m, respectively [104]. By contrast, energy costs for walking are less constant and, in turn, influenced by terrain conditions (Table 2), increasing up to four-fold when an animals walks waterlogged or soft grounds.

According to Dijkman and Lawrence [108], Bunaji bulls spent 1.47 J/m per kg to walk on firm short grass and 8.58 J/m per kg when they walked on a wet-ploughed rice field with the mud up to their knees. Fall et al. [100] recorded values of 1.59 and 2.15 per kg for oxen walking on unploughed and ploughed sandy soil, respectively. Moreover, steadily walking for a longer period imposes to animals more energy requirements than walking hard for a short period [71]. For example, a ploughing animal requires less energy than one towing a loaded cart, although ploughing needs more draught force due to the greater distance covered by the carting animal daily [108]. Similarly, when planting, the animal walking within the furrow can spend up to 20% more energy than the animal walking on the undisturbed, unplanned ground [31].

### 7.2. Feed Intake

A large body of literature has accumulated over the years around the effects of working on feed intake. Light to medium exercise has been shown to increase voluntary intake in rats and horses [78]. However, for draught animals, conflicting results are available. For instance, in oxen Pearson et al. [109] observed that over a 7-week working period exercise did not increase food intake. Conversely, Winugroho [110] observed a 25% feed intake increase in buffaloes pulling an 85 kg sled over a 39-day period. In general, over short period of times, similar or reduced feed intake are observed in non-ruminant and ruminant draught animals, suggesting that a short-term increased physical effort neither affects appetite nor increases feed intake [73]. In particular, the stress of working, especially in hot weather conditions, may lower feed intake to prevent the production of the extra-heat related to digestion [111]. Numerous studies showed reduced or similar feed intakes in draught ruminants on working days as compared with the same animals in non-working days, whereas after a working period feed intake is generally higher [74,75,112,113]. These results suggest that working ruminants may perform a compensatory intake to counterbalance the weight loss occurred during work [114]. Alternatively, ruminants may slowly adapt to the working conditions by changing their behaviour and compensating for the reduced time available for feeding [73]. In a feeding behaviour study, Pearson and Smith [115] showed that both draught cattle and buffalo increased the ingestion rate to compensate the reduced time of access to feed rather than changing the time spent feeding, as well as, with the same available time, animals spent more time ruminating at late than middle of the day.

Conversely, milking cows may increase feed intake in response to work even if the feed encumbrance is high as in the case of hay-based diets. In Ethiopia working cows showed higher intakes of natural pasture hay (72 MJ of metabolizable energy/d at 90 d postpartum) than non-working cows (60 MJ of metabolizable energy/d at 90 d postpartum) [116]. However, it was unclear whether the increased intake was concentrated soon after working or diluted over a longer period of time. Accordingly, Gemeda et al. [86] observed that dry-matter intake was higher in working compared with non-working dairy cows monitored over a 2-year period. Therefore, it has been hypothesized that, as also observed in response to lactation, the long-term response to work implies increased levels of intake as a consequence of the adaption of the cows to energy extra requirements [73]. In particular, body reserves may be preferentially used in the short term, whereas adaptive mechanisms, such as increased digestive ability and the consequential increment of dry matter intake, prevail in the long term [114].

### 7.3. Digestion

Conflicting results are available on the effect of working on digestive functions, particularly in cattle [112,114,117], whereas working buffaloes and horses tend to show an increase of digestibility [48,110,118,119]. Several studies showed that light exercise may beneficially affect the digestive functions of working animals due to a more thorough mixing of the rumen content, which, in turn, may promote microbial activity [120,121,122]. Conversely, intense working may have detrimental effects due to a shift of blood supply from the digestive tract to muscles and other peripheral tissues [123]. In addition, frequent feeding may favour rumen functions, whereas infrequent feeding (e.g., once a day) may reduce the nutrient supply to the fermenting micro-organisms, hence causing their lysis and a consequent lower rumen digestive ability [73]. Long period of working may reduce the frequency of feeding, thus, causing similar effects [48]. The feeding requirements of draught buffaloes in terms of dry matter are higher than those of cattle [108]. However, they can digest and effectively use more fibrous feeds, including by-products (e.g., rice straw, maize stubble, sugar cane bagasse, etc.), and regularly consume grass species available near the cultivated plots or in the forests [124] with an energy utilisation efficiency higher than that of other draught animals. In India, water buffaloes provide about 30% of the energy used in agriculture, at least partly because they are more efficient in the use and transformation of feed than other draught animals [23,125].

## 8. Conclusions

Over time, the use of animals for traction and transport has diminished in importance in rural environments, especially in flat areas where intensive agricultural systems have developed, systems that require other energy sources and huge inputs. These industrialised systems based on intensive livestock and forest exploitation have a marked impact on the environment in terms of GHG emission and land occupation. Under these circumstances, interest in draught animals is increasing in mountainous areas and small-scale production units for such labours as traction and transport, and as valuable sources of food. Therefore, due to the energy crisis, the lack of resources and, in some cases, the tendency to adopt agro-ecological production models aiming at reducing the impact of agricultural practices on the environment, the use of working animals in certain regions should be promoted. This would allow to obtain profits from otherwise non-usable lands in a sustainable manner, as the amount of non-renewable fuels used in agriculture would be reduced along with the emission of GHG.

The labours that these animals perform for humans are not, however, in balance with their quality of life. Although the research available on this topic is scarce, high incidences of cutaneous lesions, diseases, and injuries to hooves and snouts are clearly documented, which results in poor welfare of the animals and reduced working efficiency. The welfare of working animals has a central role in promoting human welfare, given the multiple economic and social functions they play, particularly in less advantaged areas. Appropriate handling procedures, adequate, well-maintained equipment, and improved veterinary care will all play a fundamental role in reducing the incidence of injuries and increasing the welfare of draught animals and their performances. Long period of intense working can also negatively impact the welfare of the animals both directly, due to exhaustion, and indirectly, due to reduced frequency of feed ingestion with consequent loss of weight and reduced working and, in the case of lactating animals, productive performances. The welfare of draught animals is also at the stake at the end of their working life. Differently from other productive categories, draught animals often reach this stage when they are fatigued or even exhausted. Therefore, they are potentially more exposed to suffer falls, blows, and slipping during transport to livestock markets and at slaughter. This implies that these animals should be included in public policies and specific regulations. For example, OIE has considered equids, but neglected other species, such as cattle and buffaloes. The inclusion of these species will allow specific health campaigns, animal censuses, and genetic conservation programmes devoted to draught animals.

## Figures and Tables

**Figure 1 animals-11-02683-f001:**
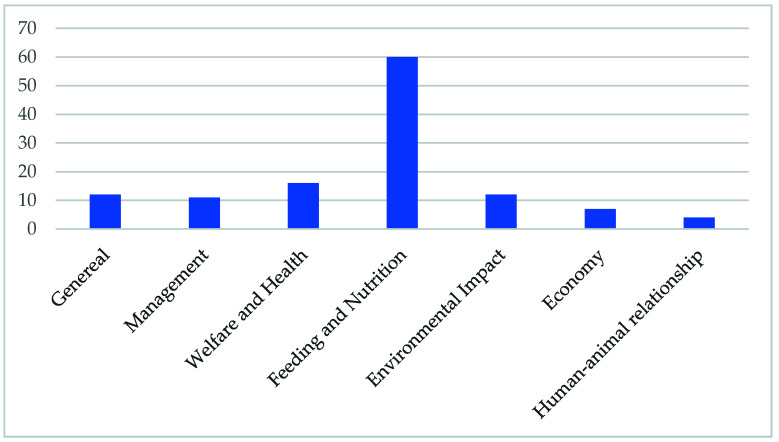
Number of publications per topic.

**Table 1 animals-11-02683-t001:** Main factors potentially impairing draught animal welfare.

Factors	Negative Effect	Preventive/Mitigation Measure	Reference
Work duration	Exhaustion, infrequent feeding, and hypoglycaemia	Appropriate work duration	Makki and Musa [44]
Heavy loads	Injuries, lameness	Loads commensurated to the animal capacity	Swann [45]
Uneven/hard surfaces	Lameness	Hoof care and appropriate trimming	Swann [45]
Harness and related gear	Sores and lesions	Maintenance of equipment and use of measures to prevent abrasion (e.g., cushions)	Makki [46]
Aversive handling	Increased fear of humans	Training of stock people	Waiblinger [47]
Poor quality feeding	Metabolic disorders, low efficiency at work	Feeding supplementation	Ffoulkes and Bamualim [48]
Transport	Injuries, lesions, fatigue, dehydration, and heat stress	Appropriate transport conditions	Minka and Ayo [49]

**Table 2 animals-11-02683-t002:** Published values of extra energy cost (ΔE, J/m per kg of body weight) above standing metabolic rate (J/kg of body weight) for draught animals walking on a dry and flat surface.

ΔE	Draught Animals	Surface Type	References
2.1	Brahman cattle	Treadmill	Lawrence and Stibbards [104]
2.1	Swamp buffalo	Treadmill	Lawrence and Stibbards [104]
2.0	Camels	Dry sandy	Rose et al. [105]
1.75	Simmental oxen	Tarmac	Rometsch [106]
1.70	Shetland ponies	Short grass	Booth [107]
1.70	Water buffalo	Concrete	Dijkman and Lawrence [108]
1.45	Zebu oxen	Laterite	Rometsch [106]
1.42	Brahman cattle	Concrete	Dijkman and Lawrence [108]
1.37	Donkeys	Gravel	Pearson et al. [102]
1.35	Brahman crossbreed steers	Concrete	Dijkman and Lawrence [108]
1.0	Zebu oxen	Laterite	Fall et al. [100]
0.97	Donkeys	Treadmill	Dijkman [103]

## Data Availability

Not applicable.

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
