# Peer review of "The Use of Draught Animals in Rural Labour"

_animals, 2021, doi:10.3390/ani11092683_

Round 1
Reviewer 1 Report
ANIMALS
Manuscript
The Use of Draught Animals in Rural Labour Including Buffaloes
This manuscript addresses an important topic related to the welfare of draught animals. The authors aimed to discuss ‘…the scientific findings on the use of draught animals […] in rural labours, including aspects concerning agricultural activities in ancient times, the relations among draught animals, farmers and rural families, the contribution of these animals to global warming, the quality of life of pack and traction animals, and their welfare conditions at the end of their working lives with a focus on buffaloes”. According to my understanding this goal is very pretentious and therefore difficult to achieve. This concern is materialized the review of "the use of animals in ancient times", which is very superficial. Thus, I suggest limiting the approach to analysing studies that focus on today's conditions and challenges (20th and 21st centuries), besides focusing on draught animals’ specific subjects.
Some paragraphs are too long, making the comprehension of the messages more difficult to the readers, please check.
Specific points that must be reviewed in the manuscript are described below.
Title
Why this emphasis on buffaloes? I suggest removing it from the title, as follow: The use of draught animals in rural labour”.
- Introduction
L79. Confuse! What do you mean for “…sustain the welfare of these animals…”? At what level? I suggest replacing "sustain the welfare of these animals" with "provide an acceptable level of welfare to these animals".
L80. I did not find the description of the principles of sustainability in the cited articles [7,11]. Please, check and correct if necessary.
L90. Remove “just” after “versus”.
L92-93. I have a feeling that you missed a citation here: Who described it as important, [13]?
L97-98. I suggest excluding the historical aspects from the manuscript, since they are not deeply explored and did not offer any valuable information about the assessment of the welfare state of the animals or their contribution to reduce global warming, for example.
L100-101. Again, it is not clear to me, why focusing on buffaloes (actually, river buffaloes, as you described in L97 and L106).
- Historical aspects and current use
L121-173. Remove all these paragraphs, following the suggestion referred to L97-98.
L191-192. Which evidence? Please, include citation(s).
- Production efficiency and economic impact
L204. Suggestion: Insert "According to the authors, the..." before "wheat yields...".
L221-222. Suggestion: excluding "and some Latin American countries". If you disagree with this, please contextualize, showing where, in Latin America, buffaloes can be characterized as "a pillar of agricultural activities".
L226-228. This sentence seems to me be in a wrong place, consider moving it to the sub-item 6.1 (L282), inserting it before "The quality of...".
L237-238. In my understanding this is an exaggerated statement, as a few draft animals do not produce enough organic matter to cover all the nutrient needs demanded by agricultural activities. The quote used to support this claim [40] is not limited to draught animals, but to an entire herd of buffaloes.
- The welfare of draught animals
L305-307. From where did you get this information? I suggest excluding this sentence, because the problems described on it are not just related to the acknowledgement of the importance of human-animal relationships, but they also depend on resources availability and sense of opportunity.
L359-364. I am not sure if this is a relevant topic to be addressed in this manuscript. The problems faced by draught animals during pre-slaughter and slaughter handling procedures, are not specific to them, affecting all animals subjected to the procedures, being them used for draught or not. Even the citation used to sustain your arguments, do not addresses this topic focusing to draught animals. I suggest removing this subject or making clear why it is addressed in the manuscript.
- Feeding draught animals
L384. This is not the best citation [59] to sustain the above statement.
L384. I suggest replacing "However, ..." with "Due to this...".
L387. I suggest starting a new sentence after "fluctuating", as follow: ". In most developing...".
L389. I suggest inserting a paragraph before "In hilly...".
L402. I suggest inserting a paragraph before "As a long...".
L428. I suggest inserting a paragraph before "In mature...".
L439. I suggest inserting a paragraph before "In particular, ...".
L476. Correct “with” instead “whit”.
L506. I suggest inserting a paragraph before "Conversely, ...".
L521. Correct “horses” instead "hourses".
- The buffalo as a pack, traction, and transport animal.
L531. I suggest removing this item, including the relevant content in other items or sub-items.
L532. Why did you not consider a more generic classification "water buffalo", which includes both, river and swamp buffaloes?
L539. Carabao is swamp buffalo!
L557. Strange statement. Is there any native grass edible for humans? Additionally, the cited authors [126] did not make any reference about native grass.
L566. Replace “river” with “water”.
L579. Insert a “.” after “[51].
L579-591. From my point of view this is not a subject that matter to this manuscript, this is because the problems reported are not specific for draught animals. I suggest removing such information.
- Perspectives and conclusions
L594. I suggest inserting "agricultural" before "systems".
L595-600. I my opinion these claims need to be substantiated with some empirical evidence. I recommend including citations that support them.
L600. Which energy crisis, could you, please, be more specific?
L617-619. Again, this seems to me to be a problem that does not only apply to draught animals, which is why I would not include this information in this manuscript.

Author Response
The Authors are willing to thank the Reviewer for the work done and the valuable suggestions which allowed to markedly improve the quality of the manuscript.
- Introduction
Reviewer comment
This manuscript addresses an important topic related to the welfare of draught animals. The authors aimed to discuss ‘…the scientific findings on the use of draught animals […] in rural labours, including aspects concerning agricultural activities in ancient times, the relations among draught animals, farmers and rural families, the contribution of these animals to global warming, the quality of life of pack and traction animals, and their welfare conditions at the end of their working lives with a focus on buffaloes”. According to my understanding this goal is very pretentious and therefore difficult to achieve. This concern is materialized the review of "the use of animals in ancient times", which is very superficial. Thus, I suggest limiting the approach to analysing studies that focus on today's conditions and challenges (20th and 21st centuries), besides focusing on draught animals’ specific subjects.
Reply from the authors
The aim has been simplified by removing the paragraph concerning the use of the animals in ancient times and the focus on buffaloes (L 96-100).
Reviewer comment
Why this emphasis on buffaloes? I suggest removing it from the title, as follow: The use of draught animals in rural labour”.
Reply from the authors
The title has been changed as suggested by the Reviewer (L 2)..
Reviewer comment
L79. Confuse! What do you mean for “…sustain the welfare of these animals…”? At what level? I suggest replacing "sustain the welfare of these animals" with "provide an acceptable level of welfare to these animals".
Reply from the authors
The statement "sustain the welfare of these animals" has been changed to "provide an acceptable level of welfare to these animals" (L 80-81).
Reviewer comment
L80. I did not find the description of the principles of sustainability in the cited articles [7,11]. Please, check and correct if necessary.
Reply from the authors
The citation has been changed to a more appropriate one (L 81).
Reviewer comment
L90. Remove “just” after “versus”
Reply from the authors
“just” after “versus” has been removed (L 91).
Reviewer comment
L92-93. I have a feeling that you missed a citation here: Who described it as important, [13]?
Reply from the authors
It has been clarified that the same authors described it as important (L 94).
Reviewer comment
L97-98. I suggest excluding the historical aspects from the manuscript, since they are not deeply explored and did not offer any valuable information about the assessment of the welfare state of the animals or their contribution to reduce global warming, for example.
Reply from the authors
The historical aspects have been omitted (L 96-100).
Reviewer comment
L100-101. Again, it is not clear to me, why focusing on buffaloes (actually, river buffaloes, as you described in L97 and L106).
Reply from the authors
The focus on buffaloes has been omitted (L 96-100).
- Historical aspects and current use
Reviewer comment
L121-173. Remove all these paragraphs, following the suggestion referred to L97-98.
Reply from the authors
All historical aspects have been omitted. Only a short initial paragraph has been left to introduce the concepts related to the current use of draught animals (L 116-124). However, the authors are willing to delete also this paragraph if the reviewer consider it unnecessary.
Reviewer comment
L191-192. Which evidence? Please, include citation(s).
Reply from the authors
A citation has been included (L 189).
- Production efficiency and economic impact
Reviewer comment
L204. Suggestion: Insert "According to the authors, the..." before "wheat yields...".
Reply from the authors
"According to the authors, the..." has been added before "wheat yields..." (L 199).
Reviewer comment
L221-222. Suggestion: excluding "and some Latin American countries". If you disagree with this, please contextualize, showing where, in Latin America, buffaloes can be characterized as "a pillar of agricultural activities".
Reply from the authors
"and some Latin American countries" has been deleted (L 216).
Reviewer comment
L226-228. This sentence seems to me be in a wrong place, consider moving it to the sub-item 6.1 (L282), inserting it before "The quality of...".
Reply from the authors
The sentence has been moved in a more appropriate paragraph, as suggested (L 279-284).
Reviewer comment
L237-238. In my understanding this is an exaggerated statement, as a few draft animals do not produce enough organic matter to cover all the nutrient needs demanded by agricultural activities. The quote used to support this claim [40] is not limited to draught animals, but to an entire herd of buffaloes.
Reply from the authors
The statement has been toned down and it is now reporting that: “…the use of draught animals may contribute to increase the amount of organic matter in the soils…” (L 227).
- The welfare of draught animals
Reviewer comment
L305-307. From where did you get this information? I suggest excluding this sentence, because the problems described on it are not just related to the acknowledgement of the importance of human-animal relationships, but they also depend on resources availability and sense of opportunity.
Reply from the authors
This sentence has been omitted (L 306).
Reviewer comment
L359-364. I am not sure if this is a relevant topic to be addressed in this manuscript. The problems faced by draught animals during pre-slaughter and slaughter handling procedures, are not specific to them, affecting all animals subjected to the procedures, being them used for draught or not. Even the citation used to sustain your arguments, do not addresses this topic focusing to draught animals. I suggest removing this subject or making clear why it is addressed in the manuscript.
Reply from the authors
It has been clarified that this topic has some specificities related to draught animals depending on the state of fatigue and exhaustion which is more likely to affect these animals at the end of their working life as compared with other productive animal categories (L 360-365). However, the authors are willing to delete this paragraph if the reviewer consider it unnecessary.
- Feeding draught animals
Reviewer comment
L384. This is not the best citation [59] to sustain the above statement.
Reply from the authors
The citation has been changed to a more appropriate reference (L 391).
Reviewer comment
L384. I suggest replacing "However, ..." with "Due to this...".
Reply from the authors
The authors make a contrast with the previous sentence: the efficiency of draught animals depends on how they are fed; notwithstanding this, they are often fed empirically. A word introducing a contrast, such as “However” seems more appropriate (L 391).
Reviewer comment
L387. I suggest starting a new sentence after "fluctuating", as follow: ". In most developing...".
Reply from the authors
The sentence has been changed as suggested (L 394).
Reviewer comment
L389. I suggest inserting a paragraph before "In hilly...".
L402. I suggest inserting a paragraph before "As a long...".
L428. I suggest inserting a paragraph before "In mature...".
L439. I suggest inserting a paragraph before "In particular, ...".
L476. Correct “with” instead “whit”.
L506. I suggest inserting a paragraph before "Conversely, ...".
L521. Correct “horses” instead "hourses".
Reply from the authors
All these minor changes have been incorporated in the text.
- The buffalo as a pack, traction, and transport animal.
Reviewer comment
L531. I suggest removing this item, including the relevant content in other items or sub-items.
Reply from the authors
This paragraph has been eliminated and relevant information included in the appropriate sections (L 137-141, 144-164, 264-271, 539-546).
Reviewer comment
L532. Why did you not consider a more generic classification "water buffalo", which includes both, river and swamp buffaloes?
Reply from the authors
“River buffalo” has been changed to “water buffalo” throughout the text.
Reviewer comment
L539. Carabao is swamp buffalo!
Reply from the authors
“River buffalo” has been changed to “water buffalo” (L 149).
Reviewer comment
L557. Strange statement. Is there any native grass edible for humans? Additionally, the cited authors [126] did not make any reference about native grass.
Reply from the authors
The statement has been changed and no reference is made about native grass and edibility (L 540-543).
Reviewer comment
L566. Replace “river” with “water”.
Reply from the authors
“River buffalo” has been changed to “water buffalo” (L 157).
Reviewer comment
L579. Insert a “.” after “[51].
Reply from the authors
This sentence has been removed.
Reviewer comment
L579-591. From my point of view this is not a subject that matter to this manuscript, this is because the problems reported are not specific for draught animals. I suggest removing such information.
Reply from the authors
This long paragraph has been markedly reduced and included in the appropriate section concerning the welfare of draught animals (L 365-369).
- Perspectives and conclusions
Reviewer comment
L594. I suggest inserting "agricultural" before "systems".
Reply from the authors
The term "agricultural" has been added before "systems" (L 549).
Reviewer comment
L595-600. In my opinion these claims need to be substantiated with some empirical evidence. I recommend including citations that support them.
Reply from the authors
The sentence has been toned down (L 550-552. No citation has been added as in the conclusion concepts already stated in the main text are reiterated and the use of references is generally discouraged.
Reviewer comment
L600. Which energy crisis, could you, please, be more specific?
Reply from the authors
This part of the sentence has been eliminated.
Reviewer comment
L617-619. Again, this seems to me to be a problem that does not only apply to draught animals, which is why I would not include this information in this manuscript.
Reply from the authors
It has been clarified that this topic has some specificities related to draught animals depending on the state of fatigue and exhaustion which is more likely to affect these animals at the end of their working life as compared with productive animals (L 573-576). However, the authors are willing to delete this paragraph if the reviewer consider it necessary.
Reviewer 2 Report
Dear Authors,
I really like your work. The review is original, very interesting and the title reflects the contents of the article. The abstract fully reflects different aspects of the study. In introduction I suggest give the number of animals with corresponding distributions on each continent.
The methods of data analysis are quite appropriate for this type of the study. The sampling technique is appropriate but I suggest indicate the number of publications taken into account, divide the publications by topic and present them graphically.
The findings of the review are presented logically with appropriate displays and explanation. The background of literature looks adequate to these review.
Other minor comments which I address to you are:
L. 149. There is a lack of logical introduction about the origin of the mules and connection to previous information about horses.
L. 152-153. Unless it is related to a generational tradition, emphasizing a social statute.
L. 312-313. Add reference.
L. 521. horses not hourses
L. 531. P. 8. Why did you focus only on buffaloes and omit the issues of other draught animals that you discussed earlier. This needs supplementation and some form of comparison.
L. 592. I suggests. Summary and conclusions
Author Response
The Authors are willing to thank the Reviewer for the work done and the valuable suggestions which allowed to markedly improve the quality of the manuscript.
Reviewer comment
I really like your work. The review is original, very interesting and the title reflects the contents of the article. The abstract fully reflects different aspects of the study. In introduction I suggest give the number of animals with corresponding distributions on each continent.
Reply from the authors
Unfortunately, these data are not available. In 1982 the Food and Agriculture Organization (FAO) of the United Nations estimated that 400 million working animals were used by 2 billion people in 30 countries. This information has been added to the text (L 65-67).
Reviewer comment
The methods of data analysis are quite appropriate for this type of the study. The sampling technique is appropriate but I suggest indicate the number of publications taken into account, divide the publications by topic and present them graphically.
Reply from the authors
The Figure 1 has been added to the text reporting the number of publications retrieved and then used per topic.
Other minor comments which I address to you are:
Reviewer comment
- 149. There is a lack of logical introduction about the origin of the mules and connection to previous information about horses.
Reply from the authors
Reviewer 1 suggested that the historical aspects be eliminated. Therefore, all the introductive paragraphs on the origin of each species have been deleted.
Reviewer comment
- 152-153. Unless it is related to a generational tradition, emphasizing a social statute.
Reply from the authors
It has been clarified that the choice of the animal species may be related to local traditions (L 131-132).
Reviewer comment
- 312-313. Add reference.
Reply from the authors
The reference has been added (L 315).
Reviewer comment
- 521. horses not hourses
Reply from the authors
“hourses” has been changed to “horses” (L 530).
Reviewer comment
- 531. P. 8. Why did you focus only on buffaloes and omit the issues of other draught animals that you discussed earlier. This needs supplementation and some form of comparison.
Reply from the authors
The focus on buffaloes has been omitted and relevant parts incorporated in the appropriate sections (L 137-141, 144-164, 264-271, 539-546).
Reviewer comment
- 592. I suggests. Summary and conclusions
Reply from the authors
This heading has been simplified to “Conclusions” (L 547).
Round 2
Reviewer 1 Report
As I wrote before, this manuscript addresses an important topic related to the welfare of draught animals. All comments and suggestions showed at the previous review were properly addressed. Please standardized the written form for British or American English spelling.